# Analysis of Synonymous Codon Usage Bias in Potato Virus M and Its Adaption to Hosts

**DOI:** 10.3390/v11080752

**Published:** 2019-08-14

**Authors:** Zhen He, Haifeng Gan, Xinyan Liang

**Affiliations:** School of Horticulture and Plant Protection, Yangzhou University, Wenhui East Road No.48, Yangzhou 225009, China

**Keywords:** *Potato virus M*, codon usage bias, pepino, natural selection, adaption evolution

## Abstract

*Potato virus M* (PVM) is a member of the genus *Carlavirus* of the family *Betaflexviridae* and causes large economic losses of nightshade crops. Several previous studies have elucidated the population structure, evolutionary timescale and adaptive evolution of PVM. However, the synonymous codon usage pattern of PVM remains unclear. In this study, we performed comprehensive analyses of the codon usage and composition of PVM based on 152 nucleotide sequences of the coat protein (CP) gene and 125 sequences of the cysteine-rich nucleic acid binding protein (NABP) gene. We observed that the PVM CP and NABP coding sequences were GC-and AU-rich, respectively, whereas U- and G-ending codons were preferred in the PVM CP and NABP coding sequences. The lower codon usage of the PVM CP and NABP coding sequences indicated a relatively stable and conserved genomic composition. Natural selection and mutation pressure shaped the codon usage patterns of PVM, with natural selection being the most important factor. The codon adaptation index (CAI) and relative codon deoptimization index (RCDI) analysis revealed that the greatest adaption of PVM was to pepino, followed by tomato and potato. Moreover, similarity Index (SiD) analysis showed that pepino had a greater impact on PVM than tomato and potato. Our study is the first attempt to evaluate the codon usage pattern of the PVM CP and NABP genes to better understand the evolutionary changes of a carlavirus.

## 1. Introduction

*Potato virus M* (PVM) is a RNA virus in the *Carlavirus* genus, *Betaflexviridae* family. PVM was first isolated from potato (*Solanum tuberosum*) in the United States in 1923 [1], and then isolated from pepino (*Solanum muricatum*), tomato and tobacco [2,3]. PVM is transmitted in a non-persistent manner by aphid and causes large economic losses of nightshade crops worldwide [2,3,4,5,6,7]. PVM contains a single stranded positive-sense RNA molecule that is approximately 8.5 kb in length and is enveloped by flexuous filamentous virions [8]. PVM possesses a cap structure at the 5′ end, a poly(A) tail at the 3′ end, and encodes six open reading frames (ORFs) [2,8,9]. ORF1 encodes a multi-domain protein that includes methyltransferase, helicase, and polymerase domains and is involved in RNA replication [10,11]. Three putative proteins within the so-called triple gene blocks (TGBs) are encoded by overlapping ORF2-4, and are involved in membrane binding, suppression of RNA silencing and cell-to-cell movement [12,13,14]. Coat protein (CP) and cysteine-rich nucleic acid binding protein (NABP) are encoded by ORF5 and ORF6, respectively [8,15].

Generally, the degeneracy of the genetic code allows for 61 triplet codons that encode all 20 amino acids such that many of them are synonymous [16,17]. Thus, codons encoding the same amino acid are termed synonymous codons. Interestingly, synonymous codons are not randomly used in a given cellular species, and the preference for specific codons over other synonymous codons by various organisms or even in different gene groups of the same genome creates a bias in codon usage. This phenomenon is known as codon usage bias (CUB) [18,19,20,21]. Several factors, such as mutation pressure, natural selection, gene length, compositional constraints, replication, selective transcription, secondary protein structure, hydrophobicity, and gene function have been shown to influence codon usage patterns [16,19,20,21,22,23,24,25,26,27]. In addition, synonymous codon usage were also biased for the capacity to form off frame stop codons [28,29,30], and codon-anticodon mismatches to compensate for tRNA misacylation [31,32,33,34]. CUB has been observed in a wide range of organisms, including viruses [18,35,36,37,38,39,40]. Compared with prokaryotic and eukaryotic genomes, the small genome size, interplay of codon usage between viruses and their hosts, and processes such as replication, protein synthesis, and transmission that depend on the host are expected to affect overall viral fitness and the avoidance of host cell responses during viral evolution.

He et al. (2019) described the evolutionary rate, timescale, host and geographical adaption of PVM on a global scale according to the coat protein and cysteine-rich NABP genes [41]. In mid-July 2019, fifteen complete genomic sequences of PVM isolates from Bangladesh, China, Czech Republic, Germany, Russia, Iran, Poland and Slovakia have been reported [2,8,9]. However, these studies reported uncertainty regarding the synonymous codon usage pattern of PVM.

In this study, we performed comprehensive analyses of codon usage and composition of PVM based on 152 nucleotide sequences of the CP gene and 125 sequences of the a cysteine-rich NABP gene to assess the evolutionary adaptation of PVM to its hosts and explored factors that may have played important roles in shaping codon usage patterns. To the best of our knowledge, our analyses provide the first insights into the codon usage patterns of a carlavirus.

## 2. Materials and Methods

### 2.1. Viral Isolates

We retrieved from GenBank 152 CP and 125 NABP CDs. The details of these PVM isolates, such as host origins, geographical locations, and collection time are provided in Appendix A.

### 2.2. Recombination and Phylogenetic Analysis

All of the PVM sequences described above were aligned using CLUSTAL X2 [42], and putative recombination sites in the aligned sequences were identified using the RDP [43], GENECONV [44], BOOTSCAN [45], MAXCHI [46], CHIMAERA [47], 3SEQ [48] and SISCAN [49] programs in the RDP4 software package [50]. RDP, GENECONV, MAXCHI, CHIMAERA, SISCAN and 3SEQ can detect when sequences are either more closely related or more distantly related in certain alignment partitions than would be expected in the absence of recombination regardless of whether they have sufficient phylogenetic support. Whereas, BOOTSCAN employs phylogeny-based comparisons of alignment partitions. The putative recombination events supported by at least three different methods (*p*-value of < 1.0 × 10^−6^) were validated and selected using default settings. The phylogenetic analysis of the PVM sequences was assessed using the neighbor-joining (NJ) method in MEGA v7 [51]. Branch support was evaluated by Kimura’s two-parameter option [52], which was used to calculate 1000 bootstrap replications for NJ analysis. The inferred trees were displayed with TreeView [53].

### 2.3. Nucleotide Composition Analysis

The component parameters of the PVM CP and NABP genes were calculated after removing five non-bias codons, such as AUG, UGG, since they are the only codons encoding Met and Trp, respectively, and termination codons (UAA, UGA, UAG). The overall nucleotide composition (A, C, T and G%) and the total AU and GC content were calculated using BioEdit. The nucleotide compositions at the third codon position of the CP and NABP genes (A3, C3, T3 and G3%) were computed using the CodonW 1.4.2 package. The GC content at the 1st, 2nd and 3rd codon positions (GC1s, GC2s, GC3s) were determined in Emboss explorer (http://www.bioinformatics.nl/emboss-explorer/), where the GC12s is the mean of GC1s and GC2s.

### 2.4. Relative Synonymous Codon Usage (RSCU) Analysis

As described by Sharp and Li (1986), the RSCU value of a codon is the ratio between the observed and expected usage frequency with the assumption that all codons for a particular amino acid are used equally [54]. An RSCU value equals 1 indicates that there is no bias for that codon, whereas codons with RSCU values of >1.6 and <0.6 are considered to be “overrepresented” and “underrepresented”, respectively. The RSCU values were calculated using the following equation:$${\mathrm{RSCU}}_{\mathrm{ij}}\;=\;\frac{g_{\mathrm{ij}}}{\sum_{j}^{\mathrm{ni}}\;g_{\mathrm{ij}}}\;\times \;\mathrm{ni}$$
where RSCU_ij_ is the relative synonymous codon usage value of the i-th codon for the j-th amino acid, and g_ij_ is the observed number of the i-th codon for the j-th amino acid that has an “n_i_” type of synonymous codon. The mean RSCU values of the PVM CP and NABP genes were calculated using MEGA 7 to determine the codon usage patterns without the effect of the amino acid composition and sequence length.

### 2.5. Principal Component Analysis (PCA)

PCA is a multivariate statistical method that was used to identify the correlations between variables and samples. Each strain was represented as a 59-dimensional vector to reduce the effect of the amino acid composition on codon usage and the RSCU value of each sense codon corresponds to each dimension. Whereas, codons UGG and AUG and the three termination codons were excluded from the analysis. PCA was analyzed using Origin 8.0.

### 2.6. Effective Number of Codons (ENC) Analysis

The ENC was used to calculate the absolute codon usage bias of the PVM CP and NABP coding sequences, regardless of the number of amino acids and gene length. The ENC values range from 20 (only one synonymous codon is used, an extreme codon usage bias) to 61 (the synonymous codons are used equally, no bias). The ENC values were calculated as follows:$$\mathrm{ENC}\;=\;2\;+\;\frac{9}{{\bar{F}}_{2}}\;+\;\frac{1}{{\bar{F}}_{3}}\;+\;\frac{5}{{\bar{F}}_{4}}\;+\;\frac{3}{{\bar{F}}_{6}}$$
where ${\bar{F}}_{k}$ (*k* = 2, 3, 4, 6) represents the mean values for *F_k_* and *k* represents k-fold degenerate amino acids. *F_k_* is estimated with the following formulae:$$F_{k}\;=\;\frac{nS\;-\;1}{n\;-\;1}$$
where *n* is the total occurrences number of the codon for that amino acid, while
$$S\;=\;\overset{k}{\underset{i=1}{\sum }}\;{\left(\frac{n_{i}}{n}\right)}^{2}$$
where *n_i_* is the total number of the i-th codon for the corresponding amino acid.

The ENC was calculated using CodonW v1.4.2. In general, the smaller the ENC value, the stronger the codon preference is. It is also accepted that ENC values ≤35 are indicative of genes with significant codon bias.

### 2.7. ENC-Plot Analysis

ENC-plot analyses (ENC values against GC3s values), which consist of plotting GC3s values in the abscissa and the ENC values in the ordinate, are used to investigate the role of mutational pressure in codon usage bias. If the only factor driving the codon usage bias is mutation pressure, these points will be on the standard curve. Otherwise, other factors such as natural selection may play a crucial role. The ENC is estimated with the following formulae:$$\mathrm{ENC}\;\mathrm{expected}\;=\;2\;+\;s\;+\;\left(\frac{29}{s^{2}\;+\;\left(1-s\right)2}\right)$$
where *s* indicates for the value of GC3s.

### 2.8. Parity Rule 2 (PR2) Analysis

Parity rule 2 (PR2) plot analysis was performed to investigate the effects of mutation and natural selection on the codon usage of individual genes by exploring the relationship of the four-codon amino acid families, with A3/(A3+U3) plotted against G3/(G3+C3). The center of the plot is where A=U and G=C (PR2), indicating a balance between mutation pressure and natural selection.

### 2.9. Neutrality Analysis

The influence of mutation bias and natural selection on codon usage were investigated by neutral plot (GC12 values against GC3 values). The mutation pressure is represented by the slope of the regression line plotted between the GC12 and GC3 contents, where regression lines that fall near the diagonal (slope = 1.0) indicates no or weak external selection pressure. In contrast, if regression curves deviate from the diagonal, it indicates a significant influence of natural selection on codon usage bias.

### 2.10. Codon Adaptation Index (CAI) Analysis

The codon adaptation index (CAI) was calculated using the CAIcal SERVER (http://genomes.urv.cat/CAIcal/RCDI/). CAI is a quantitative measure that predicts the highest relative adaptation of the viruses to their potential host. CAI values range from 0 to 1, and sequences with higher CAIs are indicative of a stronger adaptability to the host.

### 2.11. Relative Codon Deoptimization Index (RCDI) Analysis

The relative codon deoptimization index (RCDI) values for the PVM CP and NABP coding sequences were calculated using the RCDI/eRCDI server (http://genomes.urv.cat/CAIcal/RCDI/) to determine the codon deoptimization trends. An RCDI value of 1 indicates that the virus follows the host codon usage pattern and displays a host-adapted codon usage pattern. In contrast, RCDI values higher than 1 indicates lower adaptability.

### 2.12. Similarity Index (SiD) Analysis

To measure the effect of the codon usage bias of the hosts on the PVM CP and NABP genes, the SiD was calculated as follows:$$R\left(A,B\right)\;=\;\frac{\sum_{i=1}^{59}\;a_{i}\;b_{i}}{\sqrt{\sum_{i=1}^{59}\;b_{i}\;{}^{2}\sum_{i=1}^{59}\;a_{i}\;{}^{2}}}\;D\left(A,B\right)\;=\;\frac{1\;-\;R\left(A,B\right)}{2}$$
$$D\left(A,B\right)=\frac{1-R\left(A,B\right)}{2}$$
where *a_i_* indicates the RSCU value of 59 synonymous codons of the PVM coding sequences, *b_i_* indicates the RSCU value of the identical codons of the potential host, and D(*A*, *B*) (SiD) indicates the potential impact of the entire codon usage of the host on the different clades of the PVM CP and NABP genes. The SiD values range from 0 to 1.0, with higher values indicating that the host has a dominant effect on the usage of codons.

### 2.13. Gravy and Aroma Statistics

The Gravy value is calculated by CodonW (v1.4.2) and indicates the effect of protein hydrophobicity on codon usage bias, ranging in value from −2 to 2. In contrast, the Aroma value measures the effect of aromatic hydrocarbon proteins on codon usage bias.

### 2.14. Statistical Analysis

The correlation analysis was performed to identify the relationships between the GC, GC3s, ENC, the first two principal component axes, Aroma and Gravy, which were calculated using Spearman’s rank correlation analysis, with an highly significant relationship (**) of *p* < 0.01 and a significant relationship (*) of 0.01 < *p* < 0.05. All statistical analyses were performed using Origin 8.0.

## 3. Results

### 3.1. Recombination and Phylogenetic Analysis

The occurrence of recombination events can influence phylogenetic and codon usage analyses at either the gene or genome levels [55,56]. Only a previous reported recombinant isolate 501 (KC129095) [41] was observed in the CP gene regions, while no obvious recombination events were detected in NABP gene. After discarding the recombinant sequences, we conducted the phylogenetic analyses using the NJ method based on above data sets. Consistent with the findings of He et al. (2019) [41], three lineages were formed based on the CP (Appendix A) and NABP (Appendix A) coding sequences. Compared with the results of He et al., in both the CP and NABP trees, two novel potato isolates from Bangladesh and one novel potato isolate from Yunnan province of China were clustered into GP2, while two novel tomato isolates from Slovakia were clustered into GP1.

### 3.2. Nucleotide Composition Analysis

The nucleotide compositions of the PVM CP and NABP coding sequences were determined to explore the potential influence of compositional constraints on codon usage. We observed that the nucleotides G and A were the most abundant in the CP coding sequences, with mean compositions of 29.86 ± 0.53% and 26.63 ± 0.54% (Appendix A), respectively, compared with C (23.56 ± 0.25%) and U (19.94 ± 0.64%). In contrast, the nucleotide composition at the third position of synonymous codons (A_3S_, C_3S_, G_3S_ and U_3S_) in the CP coding sequences were significantly different from the nucleotide composition. The most frequent nucleotide was G_3S_ (40.25% ± 0.020), followed by U_3S_ (29.65% ± 0.018), A_3S_ (27.44% ± 0.019) and C_3S_ (26.61% ± 0.010). In contrast, the nucleotides U and G were most abundant in the NABP coding sequences, with mean compositions of 30.27 ± 0.74% and 27.75 ± 2.47% (Appendix A), respectively, followed by A (23.83 ± 1.09%) and C (18.52 ± 1.67%). In addition, the mean U_3S_ (48.91% ± 0.020) and G_3S_ (30.33% ± 0.015) values were also higher compared with A_3S_ (26.13% ± 0.018) and C_3S_ (15.93% ± 0.017) in the NABP gene sequences (Appendix A). The GC composition was higher than that of AU in the CP coding sequences, with 53.43 ± 0.58% observed compared with 46.57 ± 0.58% (Appendix A), respectively, indicating that there is a GC-biased composition of PVM CP coding sequences. Additionally, the average GC contents at the first, second, and third positions (for GC12s and GC3s) of the CP coding sequences were 52.69 ± 0.45% and 54.89 ± 1.24%, respectively. However, the AU contents (54.04 ± 1.84%) were significantly higher than that of GC (45.86 ± 1.32%) in the NABP coding sequences (Appendix A), indicating that the PVM NABP coding sequences were AU-rich. Furthermore, the mean GC12s and GC3s of the NABP coding sequences were 50.07 ± 1.09% and 37.44 ± 2.31%, respectively.

### 3.3. U- and G-Ending Codons Are Preferred in PVM CP and NABP Coding Sequences

To estimate the codon usage pattern of the PVM CP and NABP coding sequences, RSCU analysis was performed (Table 1). In the CP gene, 12 out of 18 preferred codons were G/U-ending and exhibited an equal distribution of G and U (G-ending: 6; U-ending: 6), while the remaining six were C/A-ending (C-ending: 4; A-ending: 2) (Table 1). This result shows that G- and U-ending codons are preferred in the PVM CP coding sequences. Within these preferred codons, irrespective of the PVM host, four had an RSCU value > 1.6, with the highest being GUG (2.41), indicative of extreme over-presentation, whereas the remaining preferred codons had RSCU values >0.6 and <1.6. No optional synonymous codons were underrepresented (RSCU < 0.6) from the PVM CP gene. In addition, we also calculated the host-specific RSCU values of the CP coding sequences and observed that G/U-ending codons were more common than C/A-ending codons.

In the NABP gene, among the 18 preferred codons, 14 were U/G-ending (U-ending: 10; G-ending: 4) and four were A/C-ending (A-ending: 2; G-ending: 2) (Table 1). This result shows that U- and G-ending codons are also preferred in the PVM NABP coding sequences. Within these preferred codons, 12 had a RSCU value >1.6, while the remaining six preferred codons that had RSCU values >0.6 and <1.6. Similar to the CP gene, no optional synonymous codons were underrepresented (RSCU < 0.6) for the NABP gene. According to the hosts, we also calculated the RSCU values of the NABP coding sequences and observed that G/U-ending codons were more common than C/A-ending codons. The three exceptions were the UUA, AUU and UAU codons (coding for L, I and Y, respectively), which only exhibited preferred use in potato isolates (Table 1). Both the CP and NABP RSCU analyses suggested that the preferred codons have been mostly influenced by compositional constraints (G and U in this case). In our RSCU analysis, several codons were differentially selected in different host plants in CP and NABP genes. For example, three codons AUA, AUC, and AUU encode amino acid I, however AUA was the most frequently used codons in potato rather than AUC in tomato and pepino in CP gene, whereas AUU was the most frequently used codons in potato rather than AUA in tomato and pepino in NABP gene.

### 3.4. Codon Usage Bias of the PVM CP and NABP Coding Sequences

The ENC values were calculated to infer the magnitude of the PVM CP and NABP codon usage bias. In general, the smaller the ENC value, the stronger the codon preference is. It is also accepted that ENC values ≤35 are indicative of genes with significant codon bias. Individually, the highest ENC values were both observed for the CP and NABP coding sequences from potato hosts (Figure 1), whereas the lowest values for the CP and NABP coding sequences were all obtained from pepino hosts (Figure 1). In addition, the mean ENC values for the overall CP and NABP genes were 54.91 ± 2.28 (Appendix A) and 49.66 ± 4.22 (Appendix A), respectively, indicating a relatively stable and conserved genomic composition with a low codon usage bias in all of the assayed PVM coding sequences. Compared with CP coding sequences, the lower ENC values for the NABP gene suggested a slightly greater codon bias than was observed for the CP gene.

### 3.5. Trends in Codon Usage Variations

PCA is a multivariate statistical method that was used to identify the correlations between variables and samples. In this study, PCA was performed to assess the synonymous codon usage variation in the CP and NABP coding sequences of PVM. The first four principal axes (axes 1–4) of the CP and NABP coding sequences accounted for 65.96 and 65.23% of the synonymous codon usage variation, respectively (Figure 2). The values of the first four axes for the CP coding sequences were 30.56, 18.03, 10.35 and 7.02% (Figure 2A), while those observed for NABP were 27.97, 16.23, 13.95 and 7.08% (Figure 2B). These values revealed that axis 1 was the major factor affecting codon usage for the CP and NABP genes. Furthermore, we explored the distribution of the CP and NABP coding sequences in different hosts based on the RSCU values on the first two axes (Figure 3). The PCA for both the CP and NABP genes showed several overlapped sites among the three different hosts, these suggesting similar codon usage trends (Figure 3). Notably, because the analysis included only eight tomato isolates of PVM, these results require further confirmation.

### 3.6. ENC-Plot Analysis

We performed an ENC-plot analysis for GC3s to study the factors influencing the codon usage bias of PVM according to the CP and NABP coding sequences. If the points fall below the expected curve, the codon usage is said to be affected by selection pressure rather than mutation pressure, whereas mutational pressure is indicated when the data points fall onto the expected curve. In both the CP and NABP coding sequence plots, the PVM isolates from different hosts and groups mostly clustered together below the expected ENC curve (Figure 4), indicating that the natural selection pressure was more important than mutation pressure in the PVM isolates. In relation to hosts, stronger natural selection on synonymous codon usages were observed in pepino-hosted CP and NABP genes. However, several PVM potato isolates fell onto the expected curve for both the CP and NABP coding sequences (Figure 4), indicating that the influence of mutation pressure was not completely absent.

### 3.7. Neutrality Plot

To determine the degree of mutational pressure and natural selection on the codon usage in PVM, we performed neutrality analyses between GC12 and GC3 for all of the sequences and grouped the results by the PVM hosts for the CP and NABP coding sequences (Figure 5). As nucleotide changes at the third position of the codon do not contribute to changes in the amino acid, these changes are indicative of a mutational pressure alone. In contrast, nucleotide changes that bring about changes in the altered amino acid lead to selection pressure. Significant positive correlations were observed between the GC12 and GC3 values (r^2^ = 0.0757, *P* = 7.76 × 10^−4^; r^2^ = 0.2328, *P* = 2.75 × 10^−8^) for the PVM CP and NABP coding sequences, respectively (Figure 5A,C). The slopes of the linear regression were 0.1009 and 0.1871 for all CP and NABP coding sequences (Figure 5A,C), demonstrating that mutation pressure accounted for 10.09 and 18.71% of the selection pressure, whereas natural selection accounted for 88.91 and 81.29%, respectively. With respect to the hosts, the slope of the linear regression for the PVM CP coding sequences from potato was 0.1749 (Figure 5B) and had a significant *p* value (*p* = 0.00156), whereas the correlations between GC12 and GC3 for pepino and tomato were not significant (*p* > 0.05), with observed slopes of −0.0159 and 0.2037 (Figure 5B), respectively. For the NABP coding sequences, significant correlations between GC12 and GC3 for pepino and tomato were observed, with slopes of 0.0769 and 0.1899 (Figure 5D), respectively, whereas the correlation for potato was not significant (*p* > 0.05), with a slope of 0.0298 (Figure 5D). Above all, natural selection was the dominant pressure driving the codon usage bias for the CP and NABP coding sequences of PVM.

### 3.8. Parity Analysis

We performed a PR2 bias plot analysis to determine whether the biased codon selection was restricted to highly biased genes. There is no bias in the selection or mutation pressure when the plot lie on the center, where both coordinates are 0.5 [19]. The results showed that U and G are more frequently used than A and C in the PVM CP and NABP coding sequences (Figure 6A,B), demonstrating that the codon usage pattern of PVM is also shaped by mutation pressure and other factors, including natural selection.

### 3.9. Natural Selection is a Major Player in Shaping PVM Codon Usage Patterns

To investigate the influence of natural selection on PVM codon usage patterns, we performed linear regression analysis between the general average hydropathicity (GRAVY) and aromaticity (ARO) values and the values for ENC, GC3S, GC, and the first two principle axes. Our correlation analysis based on the CP coding sequences indicated that GRAVY is significantly positively correlated with ENC but negatively correlated with axis 2. In addition, ARO showed a significant positive correlation with ENC and axis 1 but was significantly negatively correlated with GC3s, GC, and axis 2 (Table 2). For the NABP coding sequences, our correlation analysis indicated that GRAVY is significant negatively correlated with ENC, GC3s, GC, and axis 1, whereas ARO showed a significant positive correlation with GC3s axes 1 and 2 (Table 2). These results indicated that the general average hydropathicity and aromaticity are linked to the codon usage variation in PVM, indicating the influence of natural selection on the codon usage pattern.

### 3.10. Codon Usage Adaptation in PVM

To assess the codon usage optimization and adaptation of PVM to its hosts, an analysis of codon adaptation index values was performed. Particular hosts are considered to be more suitable for isolates with higher CAI values than lower values. The mean CAI values of the CP coding sequences were 0.652, 0.628, and 0.617 for the pepino, tomato, and potato hosts, respectively, while those of the NABP coding sequences were 0.721, 0.707 and 0.680 for the pepino, tomato, and potato hosts, respectively (Figure 7A). These values indicated that PVM host adaptation was greatest for pepino and the lowest for potato. Furthermore, we performed relative codon deoptimization index (RCDI) analysis to assess the cumulative effects of codon biases on gene expression. The mean RCDI values were highest for potato, followed by tomato and pepino (Figure 7B), indicating that codon usage deoptimization was the highest for the potato. Additionally, SiD analysis was also performed to understand how the PVM codon usage pattern is affected by the three hosts’ codon usage patterns (Figure 8). We observed that the SiD value of pepino was higher than that of tomato and potato in both the CP and NABP coding sequences, indicating that pepino had a greater impact on the virus than tomato and potato. 

## 4. Discussion

The codon usage pattern of viruses and hosts reflects the occurrence of evolutionary changes, such as evasion from the host’s immune system, survival, adaption [38,39,57,58,59]. Codon usage plays a significant role in the evolution of animal and human viruses, whereas our understanding of that in the evolution of plant viruses is limited. Until now, the codon usage pattern of citrus tristeza virus (CTV) [60], papaya ringspot virus (PRSV) [61], rice stripe virus (RSV) [62] and begomoviruses [63] had been reported. Low codon usage bias and higher genomic stability were observed from CTV, PRSV and RSV [60,61,62]. PVM, a carlavirus belonging to the family of *Betaflexviridae*, is an economically important pathogen of potato and pepino worldwide [3,4,5,6]. In the present study, comprehensive analyses of codon usage of PVM based on the CP and NABP coding sequences were performed. We observed that (1) the PVM CP coding sequences were GC-rich, while the NABP coding sequences were AU-rich; (2) U- and G-ending codons were preferred in the PVM CP and NABP coding sequences; (3) a relatively stable and conserved genomic composition with a low codon usage bias was observed in the PVM CP and NABP coding sequences; (4) natural selection and mutation pressure shaped the codon usage patterns of the PVM CP and NABP gene, with natural selection being the most important factor; (5) both CAI and RCDI analyses revealed that the greatest adaption of PVM was to pepino, followed by tomato and potato; and (6) pepino had a greater impact on PVM than tomato and potato.

Jenkins and Holmes (2003) reported that the overall nucleotide composition can influence codon usage bias [64]. Our nucleotide composition results showed that the PVM CP and NABP coding sequences were GC and AU-rich, respectively. In general, a GC- or AU-rich composition tends to correlate with the RSCU patterns for several organisms, including viruses [39,57,59]. For example, AU-rich genomes tend to contain codons ending with A and U, whereas GC-rich genomes tend to contain codons ending with G and C. This trend supports the influence of mutation pressure. However, we observed that U- and G-ending codons are preferred in the PVM CP and NABP coding sequences despite a higher percentage of GC versus AU, similar to the RSCU patterns observed for the Zika virus [57]. Nucleotide composition and RSCU analyses showed that selection of the preferred codons in PVM was primarily influenced by compositional constraints (G and U in this case), which accounts for the presence of mutation pressure. 

We observed that the ENC values of the PVM CP and NABP coding sequences were higher than 35 (Appendix A), indicative of a low degree of preference. Thus, we considered that the lower codon usage pattern in PVM could aid in facilitating infectivity in multiple hosts. Additionally, the strains isolated from pepino had a lower codon usage bias compared to tomato and potato. Moreover, this CAI analysis further confirmed that PVM is more adapted to pepino than tomato or potato.

Normally, the balance of mutation pressure and natural selection plays a significant role in the codon usage in eukaryotes and prokaryotes [35,39,65,66]. The results of the ENC-plot, neutrality plot and PR2 analyses clearly showed that PVM is influenced by natural selection and mutation pressure to variable degrees. Additionally, ENC-plot, neutrality plot and linear regression analysis between GRAVY and ARO values and those of the ENC, GC3S, GC, and first two principle axes demonstrated that natural selection is the major factor that has contributed to shaping codon usage of PVM. 

The emergence, dynamics, and evolution of infectious diseases can be influenced by host-parasite interactions [67,68,69,70]. PVM was isolated from pepino and tomato early in this decade but was firstly isolated from potato in the early 1920s [1,2,3]. Our previous results showed that the diversification of PVM in potato is more diverse than pepino and tomato [41], and PVM pepino isolates appear to be experiencing a new expansion [3]. In the present study, the results of our CAI analysis also showed that the PVM CP and NABP genes were strongly adapted to pepino. Moreover, RCDI analysis also supported that highest codon usage deoptimization occurred for isolates for potato, followed by tomato and pepino. These results were consistent with those of the CAI analysis, because a low RCDI may indicate strong adaptation to a host [71]. Furthermore, the SiD value for isolates from pepino was higher than those observed for tomato and potato, both in the CP and NABP coding sequences, indicating that the selection pressure of pepino on PVM was greater than that of tomato and potato, in agreement with the neutrality and ENC-plot analysis results. Therefore, a strong link between PVM and pepino was observed in this study, although potato has always been suggested to be the primary PVM host [1]. 

In summary, in this study, the codon usage patterns of the PVM CP and NABP genes were studied for the first time to better understand the evolutionary changes of a carlavirus. Results from this study promote a better understanding of the evolutionary changes of PVM, which could assist in the prevention and control of this virus.

## Figures and Tables

**Figure 1 viruses-11-00752-f001:**
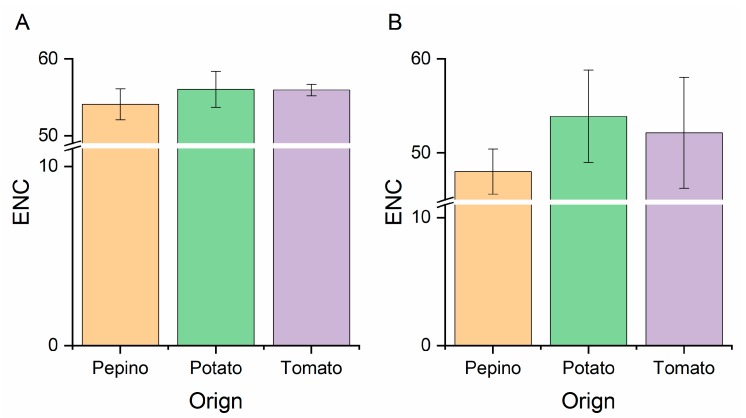
Effective Number of Codons (ENC) values of the CP (**A**) and NABP (**B**) genes of PVM from different hosts. The pepino, potato and tomato hosts are represented in light orange, light green and light purple, respectively.

**Figure 2 viruses-11-00752-f002:**
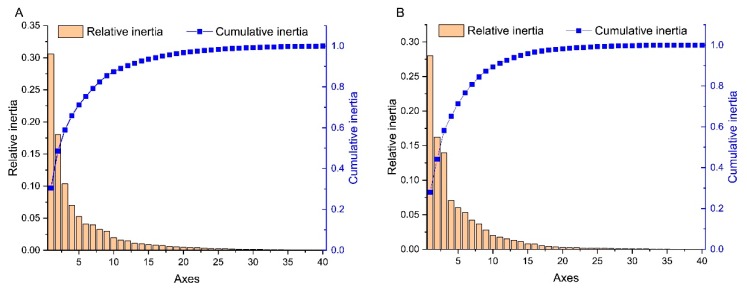
The relative and cumulative inertia of the 40 axes from a PCA of the RSCU values based on the CP (**A**) and NABP (**B**) genes of PVM.

**Figure 3 viruses-11-00752-f003:**
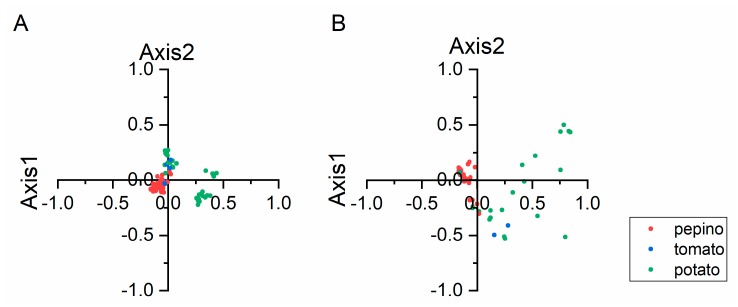
PCA based on the RSCU values of the 59 synonymous codons the CP (**A**) and NABP (**B**) genes of PVM. The pepino, potato and tomato hosts are represented in red, green and blue, respectively.

**Figure 4 viruses-11-00752-f004:**
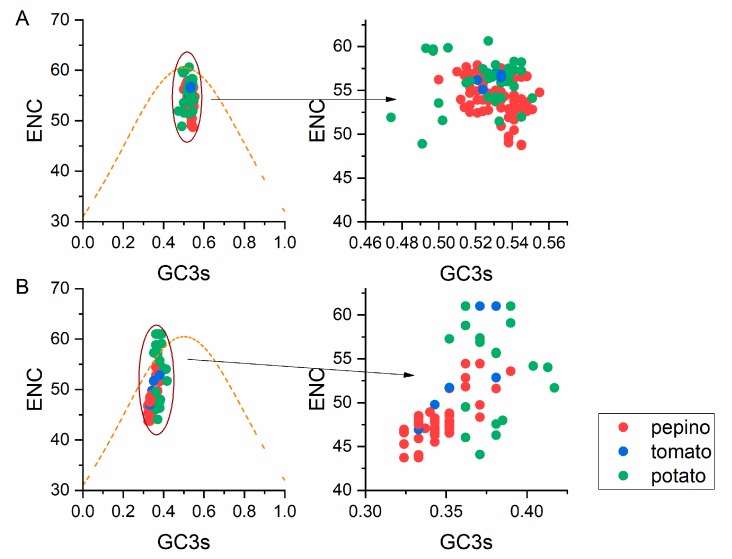
ENC-plot analysis of the CP (**A**) and NABP (**B**) genes of PVM, with ENC plotted against GC3s of different hosts. The orange line represents the standard curve when the codon usage bias is determined by the GC3s composition only. The pepino, potato and tomato hosts are represented by red, green and blue dots, respectively.

**Figure 5 viruses-11-00752-f005:**
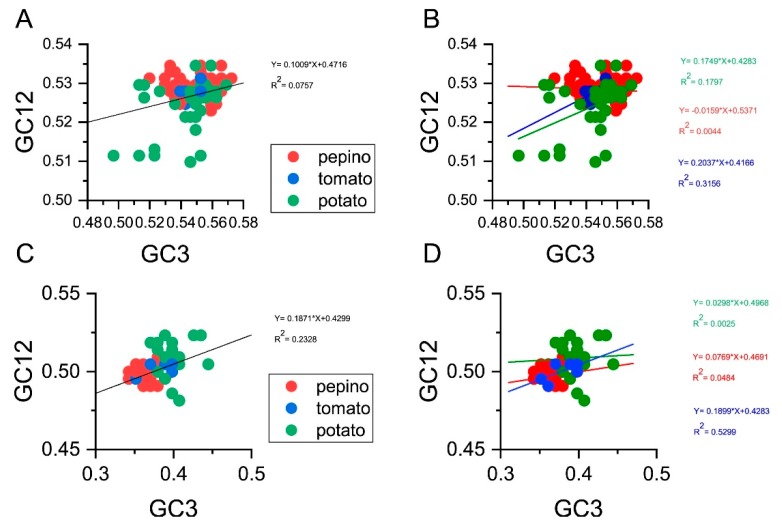
Neutrality plot analysis of GC3s against GC12s for all the coding sequences (**A**) and different hosts (**B**) based on the CP gene, and that for the coding sequences (**C**) and different hosts (**D**) based on the NABP genes, respectively. The pepino, potato and tomato hosts are represented in red, green and blue dots, respectively.

**Figure 6 viruses-11-00752-f006:**
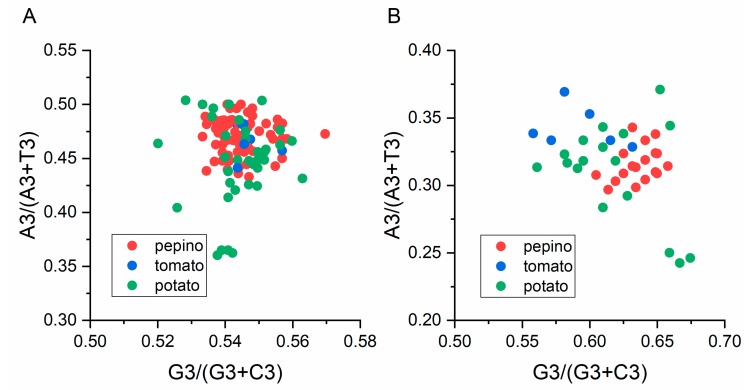
The AT [A3%/(A3% + T3%)] and GC [G3%/(G3% + C3%)] bias of the CP (**A**) and NABP (**B**) genes of PVM are shown. The center of the plot (both the coordinates is 0.5), indicating a position where there is no bias. The pepino, potato and tomato hosts are represented by red, green and blue dots, respectively.

**Figure 7 viruses-11-00752-f007:**
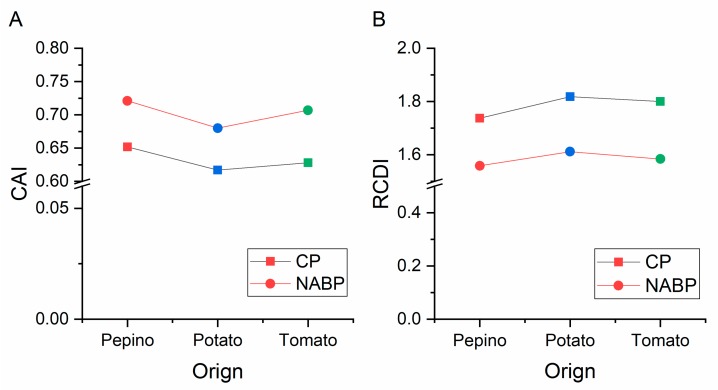
The codon adaptation index (CAI) analysis (**A**) and relative codon deoptimization index RCDI analysis (**B**) of the CP and NABP genes in relation to the natural hosts. The *x* axis represents the sequences identified in different hosts.

**Figure 8 viruses-11-00752-f008:**
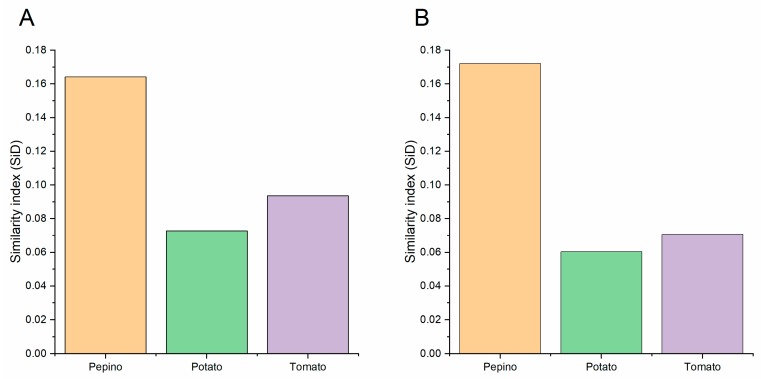
The similarity index (SiD) analysis of the PVM CP (**A**) and NABP (**B**) genes in relation to the natural hosts synonymous codon usages. The pepino, potato and tomato hosts are represented in light orange, light green and light purple, respectively. The *x* axis represents the sequences identified in different hosts.

**Table 1 viruses-11-00752-t001:** The relative synonymous codon usage (RSCU) value of 59 codons encoding 18 amino acids according to hosts of potato virus M coat protein (PVM CP) and cysteine-rich nucleic acid binding protein (NABP) genes.

Codon	aa	CP				NABP			
		Potato	Tomato	Pepino	All	Potato	Tomato	Pepino	All
UUU	F	0.89	0.63	0.65	0.73	**1.98**	**1.8**	**1.96**	**1.96**
UUC	F	**1.11** *	**1.38**	**1.35**	**1.27**	0.02	0.2	0.04	0.04
UUA	L	0.36	0.26	0.02	0.15	1.7	1.6	1.96	1.84
UUG	L	1.18	0.93	0.96	1.04	0.81	0.8	1.84	1.4
CUU	L	0.48	0.26	0.29	0.36	1.63	**2.4**	**2.05**	**1.93**
CUC	L	0.89	1.03	1	0.96	0.52	0.2	0	0.2
CUA	L	1	1.12	1.38	1.23	0.63	0.4	0.02	0.27
CUG	L	**2.1**	**2.41**	**2.36**	**2.27**	0.7	0.6	0.12	0.37
AUU	I	0.64	0.52	0.13	0.33	**1.14**	1.07	1.1	1.1
AUC	I	1.17	**1.47**	**1.74**	**1.53**	1.05	0.67	0.65	0.72
AUA	I	**1.18**	1.01	1.13	1.14	0.81	**1.26**	**1.25**	**1.18**
GUU	V	0.54	0.42	0.48	0.5	0.09	0	0	0.02
GUC	V	0.63	0.54	0.5	0.55	0.52	0.4	0.4	0.42
GUA	V	0.5	0.81	0.55	0.54	1.03	0.8	0.82	0.87
GUG	V	**2.33**	**2.24**	**2.47**	**2.41**	**2.36**	**2.8**	**2.77**	**2.69**
UCU	S	0.83	0.81	0.72	0.76	**1.83**	**2.04**	**2.38**	**2.23**
UCC	S	0.76	0.72	0.71	0.73	0.9	0.64	0.27	0.44
UCA	S	0.88	0.94	1.07	0.99	1.4	1.61	1.58	1.54
UCG	S	1.53	1.48	1.37	1.44	0.23	0	0.01	0.06
AGU	S	0.31	0.4	0.62	0.49	**1.6**	**1.71**	**1.77**	**1.72**
AGC	S	**1.7**	**1.66**	**1.51**	**1.59**	0.03	0	0	0.01
CCU	P	0.7	0.55	0.69	0.69	**2.02**	**1.7**	**1.68**	**1.75**
CCC	P	0.56	0.58	0.43	0.49	0.62	0.8	0.78	0.75
CCA	P	1.24	1.39	**1.59**	**1.45**	1.18	1.5	1.51	1.44
CCG	P	**1.51**	**1.47**	1.28	1.38	0.19	0	0.03	0.06
ACU	T	**1.85**	**1.99**	**1.95**	**1.92**	**2.68**	**2.76**	**3.05**	**2.95**
ACC	T	0.42	0.24	0.37	0.37	0.93	1.16	0.95	0.97
ACA	T	1.01	0.97	0.8	0.88	0.19	0	0	0.04
ACG	T	0.72	0.8	0.89	0.83	0.19	0.07	0	0.04
GCU	A	**1.51**	**1.57**	**1.34**	**1.42**	**2.05**	**2.38**	**2.3**	**2.26**
GCC	A	0.86	0.78	0.94	0.9	0.05	0.05	0.14	0.12
GCA	A	1.03	1.03	1.17	1.11	0.48	0.66	0.41	0.44
GCG	A	0.6	0.62	0.54	0.57	1.43	0.91	1.15	1.19
UAU	Y	0.99	**1.08**	**1.01**	**1.01**	**1.13**	0.86	0.84	0.91
UAC	Y	**1.01**	0.92	0.99	0.99	0.87	**1.14**	**1.16**	**1.09**
CAU	H	0.77	0.4	0.46	0.56	0.88	0.53	0.72	0.74
CAC	H	**1.23**	**1.6**	**1.54**	**1.44**	**1.12**	**1.47**	**1.28**	**1.26**
CAA	Q	0.74	0.58	0.44	0.56	**1.37**	**2**	**2**	**1.89**
CAG	Q	**1.26**	**1.42**	**1.56**	**1.44**	0.63	0	0	0.11
AAU	N	**1.18**	**1.34**	**1.29**	**1.25**	**1.64**	**1.33**	**1.34**	**1.4**
AAC	N	0.82	0.66	0.71	0.75	0.36	0.67	0.66	0.6
AAA	K	0.76	0.75	0.62	0.68	0.31	0.53	0.53	0.49
AAG	K	**1.24**	**1.25**	**1.38**	**1.32**	**1.69**	**1.47**	**1.47**	**1.51**
GAU	D	**1.24**	**1.32**	**1.45**	**1.37**	**1.97**	**1.78**	**2**	**1.98**
GAC	D	0.76	0.68	0.55	0.63	0.03	0.22	0	0.02
GAA	E	0.7	0.68	0.66	0.68	0.73	0.8	0.78	0.77
GAG	E	**1.3**	**1.32**	**1.34**	**1.32**	**1.27**	**1.2**	**1.22**	**1.23**
UGU	C	0.72	0.83	0.71	0.72	**1.58**	**1.62**	**1.66**	**1.64**
UGC	C	**1.28**	**1.17**	**1.29**	**1.28**	0.42	0.38	0.34	0.36
CGU	R	0.65	0.31	0.26	0.4	1.16	1.37	1.37	1.32
CGC	R	1.11	0.81	0.78	0.9	1.07	0.91	0.91	0.95
CGA	R	0.93	1.47	1.5	1.3	0.59	0.74	0.52	0.56
CGG	R	0.53	0.56	0.53	0.53	0.16	0	0	0.03
AGA	R	0.82	0.69	0.69	0.74	1.3	1.26	1.34	1.32
AGG	R	**1.96**	**2.16**	**2.24**	**2.14**	**1.73**	**1.71**	**1.85**	**1.82**
GGU	G	**1.28**	1.01	**1.15**	**1.19**	**2.09**	**2.57**	**2.85**	**2.66**
GGC	G	0.8	0.95	0.98	0.92	0.85	0.14	0.01	0.21
GGA	G	0.9	1.01	1	0.96	0.75	0.71	0.57	0.62
GGG	G	1.02	**1.03**	0.86	0.93	0.31	0.57	0.57	0.51

* The most frequently used codons are shown in bold.

**Table 2 viruses-11-00752-t002:** Correlation analysis among GRAVY, ARO, ENC, GC3_S_, GC, and the first two principle component axes.

Gene		ENC		GC3s		GC		Axis1		Axis2	
		r	P	r	P	r	P	r	P	r	P
CP	Gravy	0.17691 *	0.0309	−0.00431 ^ns^	0.95835	−0.01278 ^ns^	0.87702	0.14168 ^ns^	0.0848	−0.29638 **	2.42 × 10^−4^
	Aromo	0.27855 **	5.82 × 10^−4^	−0.26752 **	9.73 × 10^−4^	−0.23363 **	0.00414	0.39558 **	5.94 × 10^−7^	−0.19631 *	0.01642
NABP	Gravy	−0.26677 **	0.00298	−0.48469 **	1.54 × 10^−8^	−0.4726 **	3.88 × 10^−8^	−0.62506 **	1.41 × 10^−14^	−0.16583 ^ns^	0.06793
	Aromo	−0.15843 ^ns^	0.08135	0.21532 *	0.01723	0.11425 ^ns^	0.21018	0.44691 **	2.47 × 10^−7^	0.30524 **	6.29 × 10^−4^

^ns^ non-significant (*p* > 0.05); * represents 0.01 < *p* < 0.05; ** represents *p* < 0.01.

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
