# Peer review of "Analysis of Synonymous Codon Usage Bias in Potato Virus M and Its Adaption to Hosts"

_viruses, 2019, doi:10.3390/v11080752_

Round 1

Reviewer 1 Report

This manuscript describes a detailed analysis of codon usage in potato virus M (PVM) isolates from different plants (pepino, tomato, potato), focusing on two protein-coding genes of this virus – capsid protein (CP) and cysteine rich nucleic acid binding protein (NABP). The overall conclusion, if I understood correctly, is that while there is an overall difference in GC content between these two genes (CP is more GC rich than NABP), the codon usage bias is low for both genes, and the codon usage pattern of this virus was shaped more by the pepino host than the other two hosts.

Overall this is a valuable study that examined PVM genome evolution by analyzing codon usage patterns. I do not understand much of the statistical methodology, thus cannot judge whether the conclusions are supported by their analysis. My major criticism lies on the writing of the manuscript. Surprisingly, with this manuscript it is not the English per se that bothers me. Rather, it is how loosely the authors used certain terms and definitions. I have quite a few examples listed below. I have the general sense that they are not sufficiently careful with the limitations of certain hypotheses, and need to be more aware of the conditions and circumstance under which certain theories apply, and refrain themselves from using terms and concepts that they do not fully understand.

They also need to try harder to improve the readability of the ms so that it appeals to a more general audience.

Ln 15, provide the full names for “CP” and “NABP”.

Ln17, what does “the lower codon usage” mean?

Ln20, what does CAI mean? And RCDI?

Ln29 and 30, do both potato and pepino have the same formal name (Solanum muricatum)?

Ln30, near end, add “manner” after “non-persistent”

Ln32, end, delete “by a”

Ln34, the ORF1 protein is NOT a polyprotein because it is not proteolytically cleaved. It is just a multi-domain protein.

Ln36, change “named” to “within the so-called”

Ln42, end, “selected” is not a proper word here. It should be “not randomly used in a given cellular species”

Ln43, delete “unequal”, change “of” to “for” after “preference”

Ln47, why would secondary protein structure affect codon usage? Do you mean the secondary structure of the mRNA?

Also I strongly doubt that “external environment would affect codon usage. This would imply that the specific sequence of a gene can be changed from generation to generation due to environmental changes. This is Lamarckian. Be careful with your statements.

Ln53-54, delete until “He et al……” This front part is misleading, and the sentence structure is awkward.

Ln62, change to “that may have played”, again be careful with your statement. Don’t be so sure with this type of analysis. They are hard to verify experimentally.

Ln176, why “also”? Please explain how this recombination was discovered? Was it between two different isolates of PVM, or between PVM and a different virus? If it is just one isolate, and particularly just one sequence (KC129095), how can you detect recombination?

Ln178, what “above datasets”? you have to explain in the results section how your dataset was assembled (how many sequences were included?).

The note under Table 1: the word “optimal” may not be appropriate. It implies a functional advantage. However, the type of analysis merely identified the most frequently used codons for a particular amino acid in a particular protein, in a particular plant species.

Ln236, “lower” should be “low”

Table 1. the word “origin” is missing the second “i”

Sections 3.4 and 3.5, brief explanations are needed for “ENC” and “PCA” as to what they mean and why are they used here. I understand that they were explained in the Methods section already, but it is always better to refresh reader’s memory.

Ln253, “eight tomato sequences” is misleading. I am guessing it meant to be eight tomato isolates of PVM.

Ln251-254, please try to rephrase the main message of this section in a way that is easily understandable to a more general audience. As it is it is unclear what the analysis in this section accomplished.

Ln352, delete “and evolution”

Ln353, it is wrong to say that “its role……. Is limited”. It is only that our understanding of that is limited, due to limited analysis.

Ln355, please described the findings from those previous analyses.

Ln361, “lower” should be “low”

Ln362, you need to better define how you use the terms “natural selection” and “mutation pressure”. To most people “mutation pressure” is part of “natural selection”, unless you can show a type “mutation pressure” that is not subject to “natural selection”, you need to modify the sentence.

Also, you need to better articulate how, for example, would the codons of the CP gene be differentially selected in different host plants, and how they would be differentially selected from the NABP gene.

Reviewer 2 Report

Review of ms Analysis of synonymous codon usage bias in potato virus M and its adaption to hosts, authors: Zhen He, Haifeng Gan and Xinyan Liang
Reviewer: Herve Seligmann

Ther manuscript describes analyses of synonymous codon usages in the PVM, Carlaviruses, using several indices to detect biases in synonymous codon usages. The main findings are:1 natural selection for synonymous codon usage occurs, and specifrically in PVMs from some, rather than other, plant hosts; 2. PVM codon usages are best adapted to pepino.
The manuscript is understandable but the language is imperfect and requires edition, see suggestions below. Some points are unclear and should be clarified in the text, probably due to errors or sentence construction.
Below are also some suggestions for functional causes from the literature for synonymous codon biases. I urge the authors to state in the manuscript that the indices used estimate statistical biases, and that natural selection is only an interpretation of these biases. They should present alternative interprettaions, and explain why, overall, the selection interpretation is valid.
I recommend a revision, which is not small, but can not be considered as major, as the overall issues are fine, analyses adequate and results clear. Analysis methodology, and results are interesting.

My edits are between { }
Potato virus M (PVM) is an{->a} RNA virus in the Carlavirus genus, Betaflexviridae family. PVM was
first isolated from potato (Solanum muricatum) in the United States in 1923 [1], and then isolated from
pepino (Solanum muricatum), tomato and tobacco [2,3]. PVM is transmitted {in a non-persistent<-delete, or incomplete, do you mean: ....is a nonpersistent virus, and is transmitted by...} by
aphid and causes large economic losses of nightshade crops worldwide [2,7]. PVM contains a single
stranded positive-sense RNA molecule that is appropriately{->approximately} 8.5 kb in length and is enveloped by a
by flexuous filamentous virions [8].

Synonymous codon usage: also biased for the capacity to form off frame stop codons: Seligmann and Pollock 2003; Seligmann 2010; Seligmann 2019
Seligmann H, Pollock DD. The ambush hypothesis: hidden stop codons prevent off-frame gene reading. DNA Cell Biol. 2004 Oct;23(10):701-5.
Seligmann H. The ambush hypothesis at the whole-organism level: Off frame, 'hidden' stops in vertebrate mitochondrial genes increase developmental stability. Comput Biol Chem. 2010 Apr;34(2):80-5
Seligmann H. Localized Context-Dependent Effects of the "Ambush" Hypothesis: More Off-Frame Stop Codons Downstream of Shifty Codons. DNA Cell Biol. 2019 Jun 3. doi: 10.1089/dna.2019.4725.

biased for codon-anticodon mismatches to compensate for tRNA misacylation: Seligmann 2010, 2011, 2012; Barthélémy and Seligmann 2016
Seligmann H. Do anticodons of misacylated tRNAs preferentially mismatch codons coding for the misloaded amino acid? BMC Mol Biol. 2010 May 28;11:41
Seligmann H. Error compensation of tRNA misacylation by codon-anticodon mismatch prevents translational amino acid misinsertion. Comput Biol Chem. 2011 Apr;35(2):81-95.
Seligmann H. Coding constraints modulate chemically spontaneous mutational replication gradients in mitochondrial genomes. Curr Genomics. 2012 Mar;13(1):37-54
Barthélémy RM, Seligmann H. Cryptic tRNAs in chaetognath mitochondrial genomes. Comput Biol Chem. 2016 Jun;62:119-32.

Also see how "synonymous" codons might not be actually synonymous, but might give information on protein secondary structure:
Seligmann H, Warthi G. Genetic Code Optimization for Cotranslational Protein Folding: Codon Directional Asymmetry Correlates with Antiparallel Betasheets, tRNA Synthetase Classes. Comput Struct Biotechnol J. 2017 Aug 12;15:412-424.

To date (May, 2019)-> now is mid-July 2019

One hundred fifty-two CP and 125 NABP coding sequences were retrieved from GenBank-> We retrieved from GenBank 152 CP and 125 NABP CDs.

2.2. Recombination and phylogenetic analysis
recombination sites: please explain the differences between the methods used by the programs. At least give broad explanations about different approaches used.

line 136 investigateed->investigated
line 169 the first two {principal component} axes
line 170 extremely->highly

3.1. Recombination and phylogenetic analysis
A previous study reported that recombination in isolate 501 from Iran (KC129095) [33] was also observed in the CP gene regions-> this sentence is unclear: did your analyses confirm this, or not? state it clearly.
Did your analyses find other recombibations? State this clearly, as it stands, one can guess you did not find any, but it is not clear.

line 198 indicating that there is an{->a} GC-rich {rich->biased} composition of PVM CP coding sequences
line 203 the mean GC12s and GC3s the of {of the, not the of} NABP coding sequences were 50.07 ± 1.09% and 37.44 ± 2.31%,

Figure 3: should be larger
Figure 4B indicates stronger natural selection on synonymous codon usages in pepino-hosted NABP genes. This should be stated and discussed. Indepepndently of the observation, a statistical test should check if this effect is significant.
Lines 288-296: tests for statistical significance between regression lines for different hosts should be done. These tests are based on multiple regressions, where additional variables (sometimes called dummy variables), encode potential effects of the hosts.
The test must be done on pairs of hosts, where data from one host is encoded "0" in the variable, and from the other host "1". Significance of the multiple regression coefficient for that variable means that the constants of the regressions for these hosts are statistically different. This will probably show that potato-hosted viruses have different constants than tomato- and pepino-hosted viruses.
A similar minded test exists to estimate differences between slopes, where a new "dummy variable" is built by multiplying the X (independent) variable by the 0/1 variable described above, and this new variable is integrated as indepepndent in the multiple regression.
One can do tests only on differences between constants, only between slopes, and tests including both dummy variables, for constant and slopes. Using only tests for the constant is a reasonable approximation, an in depth analysis would tests all 3 options: only constant, only slope, and constant+slope. Again, this must be done for pairs of hosts, hence pepino-tomato; pepino-potato, and tomato-potato, separately for each gene.

Figure 5. Same point on differences between regression coefficients. I do not understand what is the difference between graphs A, B, C and D, please explain better in figure legend.

Lines 326-327 prinicple {component} axes
line 331 particular hosts is {->are}
line 338 The mean RCDI values was {->were} highest for

3.10. Codon usage adaptation in PVM
figure 8 and related text: a statistical test for significances of differences between SID for different hosts in each gene is missing

Line 347: in relation to the natural hosts {synonymous codon usages}
line 366 Jenkins et al. (2003)->reference format?
line 370 tends->tend
line 371 support->supports
line 404 This results of this study->Results from this study....

Round 2

Reviewer 1 Report

the authors have addressed most of my concerns. I do not object its acceptance.

Author Response

Response to Reviewer #1

the authors have addressed most of my concerns. I do not object its acceptance.

Response: We thank the reviewer for the fine comments.

Reviewer 2 Report

seems acceptable for publication, after correcting small edits found in the new (highligthed text), described below:

Line

48 addation->addition

56 change "Now is mid-July 2019" to "In mid-July 2019"

278 "in aspect of" -> In relation to

418 Results from this study promotes->promote

Author Response

Response to Reviewer #2

seems acceptable for publication, after correcting small edits found in the new (highligthed text), described below:

Response: We thank the reviewer for these fine comments. We provided a point-by-point response to the comments as following.

Line

48 addation->addition

Response: Thanks. Modified as suggested.

56 change "Now is mid-July 2019" to "In mid-July 2019"

Response: Thanks. Modified as suggested.

278 "in aspect of" -> In relation to

Response: Thanks. Modified as suggested.

418 Results from this study promotes->promote

Response: Thanks. Modified as suggested.